# Photocatalytic and Oxidative Synthetic Pathways for Highly Efficient PANI-TiO_2_ Nanocomposites as Organic and Inorganic Pollutant Sorbents

**DOI:** 10.3390/nano10030441

**Published:** 2020-02-29

**Authors:** Carolina Cionti, Cristina Della Pina, Daniela Meroni, Ermelinda Falletta, Silvia Ardizzone

**Affiliations:** 1Department of Chemistry, Università degli Studi di Milano, via Golgi 19, 20133 Milano, Italy; carolina.cionti@unimi.it (C.C.); cristina.dellapina@unimi.it (C.D.P.); silvia.ardizzone@unimi.it (S.A.); 2Consorzio Interuniversitario Nazionale per la Scienza e Tecnologia dei Materiali (INSTM), via Giusti 9, 50121 Florence, Italy; 3ISTM-CNR, via Golgi 19, 20133 Milano, Italy

**Keywords:** nanocomposites, hybrid nanoparticles, polyaniline, titanium dioxide, pollutant sorption, green synthesis

## Abstract

Polyaniline (PANI)-materials have recently been proposed for environmental remediation applications thanks to PANI stability and sorption properties. As an alternative to conventional PANI oxidative syntheses, which involve toxic carcinogenic compounds, an eco-friendly procedure was here adopted starting from benign reactants (aniline-dimer and H_2_O_2_) and initiated by ultraviolet (UV)-irradiated TiO_2_. To unlock the full potential of this procedure, we investigated the roles of TiO_2_ and H_2_O_2_ in the nanocomposites synthesis, with the aim of tailoring the properties of the final material to the desired application. The nanocomposites prepared by varying the TiO_2_:H_2_O_2_:aniline-dimer molar ratios were characterized for their thermal, optical, morphological, structural and surface properties. The reaction mechanism was investigated via mass analyses and X-ray photoelectron spectroscopy. The nanocomposites were tested on both methyl orange and hexavalent chromium removal. A fast dye-sorption was achieved also in the presence of interferents and the recovery of the dye was obtained upon eco-friendly conditions. An efficient Cr(VI) abatement was obtained also after consecutive tests and without any regeneration treatment. The fine understanding of the reaction mechanism allowed us to interpret the pollutant-removal performances of the different materials, leading to tailored nanocomposites in terms of maximum sorption and reduction capability upon consecutive tests even in simulated drinking water.

## 1. Introduction

Polyaniline (PANI) is one of the most appealing members of the intrinsically conductive polymer family thanks to its unique features, such as high stability, acid-base characteristics and peculiar redox properties. For all these reasons, PANI-based materials and composites are now a hot-topic in the materials science community and have been applied in numerous fields, ranging from photovoltaics, anticorrosion and biomedical devices to sensors, electrochemistry and electronics [1,2,3,4,5,6,7,8,9,10,11]. Furthermore, in recent years PANI has drawn increasing attention as alternative sorbent for pollutant remediation from both water and air due to its high sorption capability and ease of regeneration [12,13,14,15,16]. In this regard, polyaniline represents a promising alternative to currently adopted materials, such as activated carbon, thanks to the fact that PANI does not require energy-consuming thermal activation and regeneration treatments.

On the other hand, PANI traditional synthesis involves noxious and toxic reagents (aniline monomer and strong oxidants such as (NH_4_)_2_S_2_O_8_), leading to carcinogenic by-products (benzidine and trans-azobenzene) and producing large amounts of inorganic waste during the product purification (mostly sulphates). Some attempts were carried out to find alternative green procedures. Among them, the enzymatic route based on the metal-enzyme Laccase is attractive but still remains a method for small-scale preparation [17]. More interestingly, the electrochemical preparation of PANI has been extensively explored. This latter method represents the best approach to produce PANI films characterized by a fine control of thickness and morphology [18]. However, electrochemical syntheses show lower production yield than chemical oxidation, do not permit the production of complex polymer-inorganic composites, and present difficulties in scale-up [18,19]. In order to achieve high production yields, long-term electropolymerization is needed: this procedure leads to the accumulation of by-products including p-benzoquinone, p-hydroquinone and co-oligomers of aniline and p-benzoquinone and acid [18]. Besides their significant side effects on the properties of the final product, the discarded electrolyte solution is highly pollutant. As an alternative, an environmentally friendly process was developed by some of us [14] adopting more benign compounds, such as aniline dimer (N-(4-aminophenyl)aniline) as starting reagent, H_2_O_2_ as oxidant and FeCl_3_ as catalyst. Even though aniline dimer is produced by chemical routes and labeled as toxic and irritant, it is not carcinogenic, differently from aniline monomer, and its solid form makes handling easier and safer. Furthermore, PANI synthesis by aniline dimer leads to H_2_O as the only co-product thus simplifying the material post-treatment and eliminating the production of carcinogenic species. However, because of the fast polymerization, this procedure does not allow a strict control over the polymer structural and morphological features and the resulting polyaniline is characterized by a compact morphology, low surface area and, as a consequence, poor sorption properties. Very recently, we developed a new green, two-step synthesis taking advantage of the photocatalytic properties of TiO_2_ semiconductor under UV irradiation [20]. The outstanding properties of the resulting PANI-TiO_2_ nanocomposite, including unique PANI crystallinity, high surface area and porosity, prompted us to further investigate these promising composite materials. 

Here, PANI-TiO_2_ nanocomposites were prepared by varying the stoichiometric amounts of the two main actors of this procedure, TiO_2_ and H_2_O_2_, with the aim of tailoring the physicochemical properties of the final material. Differences among the various products were highlighted by an all-round characterization including structural, morphological, thermal, and optical analyses. Moreover, we took advantage of the unique opportunity offered by the present synthetic approach, in which the oligomerization and polymerization steps are clearly separated, to investigate the early stages of PANI synthesis. Samples were characterized after the oligomerization step by a combination of spectroscopic and spectrometric techniques to disentangle the photocatalytic and catalytic TiO_2_ contributions. Finally, we demonstrate the nanocomposites ability to remove both organic (methyl orange) and inorganic (Cr(VI)) pollutants in water and in a wide concentration range; the materials reusability was also evaluated without any regeneration treatment, also in conditions closer to real life usage. The fine understanding of the reaction mechanism allowed us to interpret the pollutant removal performance of the different materials, leading to tailored nanocomposites in terms of maximum sorption capability upon consecutive tests even in simulated drinking water.

## 2. Materials and Methods 

Sigma Aldrich (Darmstadt, Germany) reagents were adopted for experimental procedures, unless specified otherwise. Suspensions and solutions were prepared with Milli-Q water.

### 2.1. Sample Preparation

PANI-TiO_2_ nanocomposites were synthesized by a two-step green method. In a typical synthesis, 1 g of aniline dimer (*N*-(4-aminophenyl)aniline) was dissolved in 290 mL of a 0.09 M HCl aqueous solution. Then, 175 mg of P25 TiO_2_ (Evonik, Bitterfeld, Germany) was added under vigorous stirring and irradiated with UV light (halogen lamp 500 W Jelosil, Vimodrone, Italy; 30 mW cm^−2^) for 135 min. Then, the suspension was stirred in the dark for 4 h at 25 °C. In the second step, 0.55 mL of 30% H_2_O_2_ aqueous solution was added and the reaction mixture was stirred for 18 h in the dark. The final product was collected by filtration and washed with 50 mL of water and few milliliters of acetone, then dried at room temperature overnight. Hence, this sample, named 1 × 1 in the following, was prepared using TiO_2_:H_2_O_2_:aniline dimer molar ratios = 0.4:1.0:1.0. 

In order to disentangle the relative contributions of TiO_2_ and H_2_O_2_, different samples were prepared by modulating the TiO_2_:H_2_O_2_:dimer molar ratios while keeping all other parameters unchanged (aniline dimer concentration, HCl amount, temperature, irradiation time). Samples were labelled as *n* × *m* where *n* is a number indicating the TiO_2_ molar amount with respect to the reference synthesis (i.e., ratio between the adopted TiO_2_ moles and the TiO_2_ moles in the 1 × 1 reference), while *m* indicates the H_2_O_2_ amount also compared to the reference sample (i.e., adopted moles of H_2_O_2_ divided by the H_2_O_2_ moles used in 1 × 1 reference). In a first series, the amounts of both TiO_2_ and H_2_O_2_ were doubled (2 × 2) and tripled (3 × 3). Another set of samples was prepared by varying only the photocatalyst quantity, doubling (2 × 1) and tripling it (3 × 1). The last series was obtained by doubling and tripling only the H_2_O_2_ amount (1 × 2 and 1 × 3, respectively). For better clarity, Table 1 lists the prepared nanocomposites, along with the reactant molar ratios.

For the sake of comparison, PANI-TiO_2_ composites were compared with a pure PANI sample (named PANI_ref) prepared according to the green synthetic approach previously proposed by some of us, using aniline dimer, H_2_O_2_ (H_2_O_2_ : dimer molar ratio = 5) and FeCl_3_ catalyst [14].

### 2.2. Materials Characterization 

X-ray powder diffraction (XRPD) analyses were carried out with a PW 3710 Bragg-Brentano goniometer (Philips, Amsterdam, The Netherlands) equipped with a scintillation counter, a slit with 1° divergence, a receiving slit of 0.2 mm and a 0.04° Soller slit system. A graphite-monochromatic Cu K*α* radiation was adopted at a nominal X-ray power of 40 kV × 40 mA. Diffractograms were obtained in a 2*θ* range between 10° and 80°.

A UV-2600 UV-vis spectrophotometer (Shimadzu, Kyoto, Japan) was employed to record UV-vis absorption spectra in the 200–1000 nm range of samples dissolved in DMF with and without HCl addition.

Fourier Transform infrared (FTIR) spectra were collected in the 400–4000 cm^−1^ range using a Spectrum 100 ATR spectrometer (PerkinElmer, Waltham, MA, USA).

The Brunauer-Emmett-Teller (BET) method was adopted to determine the composite specific surface area from adsorption isotherms of N_2_ in subcritical conditions, measured using a SA3100 instrument (Beckman-Coulter, Brea, CA, USA).

A LEO 1430 instrument (Zeiss, Jena, Germany) was employed to acquire Scanning Electron Microscopy (SEM) images. 

Thermo gravimetric analyses (TGA) were performed in air (5 °C min^-1^, temperature range 30–900 °C) with a TGA/DSC 3+ instrument (Mettler Toledo, Switzerland) equipped with a 70 μL alumina crucible.

An M-probe apparatus (Surface Science Instruments, Pasadena, CA, USA) was adopted for X-ray photoelectron spectroscopy (XPS) analyses: the source was a monochromatic Al K*α* radiation (1486.6 eV). The accuracy of the reported binding energies (B.E.) can be estimated as ± 0.2 eV. Peak fitting was carried out using combined Gaussian−Lorentzian curves, upon background correction with the Shirley method. Binding energies were corrected for specimen charging by referencing the Ti(IV) component of the Ti 2p_3/2_ peak at 458.5 eV.

Electrospray ionization spectrometry (ESI-MS) analyses were conducted on an ion trap mass spectrometer (LCQ Fleet, Thermo Scientific, Waltham, MA, USA) operating in the positive ion mode. Aliquots were directly infused into the ESI source at a flow rate of 20 μL min^−1^ by a micro syringe. The ESI source conditions were as follows: heated capillary temperature 295 °C; sheath gas (N_2_) flow rate 4 L min^−1^; spray voltage 5.5 kV; tube lens offset voltage 95 V. The m/z range employed in all experiments was 50–2000. 

### 2.3. Study Of The Reaction First Step

In order to investigate the first stage of the reaction, two tests were performed by stopping the reaction before the addition of H_2_O_2_, in order to characterize the formed oligomers via FTIR and XPS analyses. In one test, by following the reference (1 × 1) synthetic procedure, TiO_2_ powder was suspended in an aqueous solution of the aniline dimer, acidified by HCl, and was irradiated with UV light for 2.25 hours. After irradiation, the reaction was stopped by centrifugation and the solid fraction was retrieved and characterized. The second test was carried out in the same way, but without UV irradiation: the suspension was kept in the dark under stirring for 2.25 hours after TiO_2_ addition and then centrifuged.

### 2.4. Dye-Removal Tests

The nanocomposites sorption capabilities were tested towards methyl orange (MO) removal in water medium without any direct irradiation. For each composite, 50 mg of sample was suspended in 20 mL of a MO solution (10–200 ppm) for 20 minutes (at spontaneous pH, around 5, and room temperature) and then recovered by centrifugation. The remaining dye concentration was determined by UV-vis spectroscopy at the wavelength of maximum absorption (504 nm). 

The influence of electrolytes was studied by performing sorption tests both in Milli-*Q* water and in simulated drinking water, prepared according to Annex B2 of the Second Protocol of the French Norm NF P41-650 regarding the Specifications for Water Filter Pitchers (Appendix A) [21]. 

Consecutive sorption tests were performed without regenerating the sample between tests: powders were simply retrieved by centrifugation at the end of the sorption step and re-suspended in MO solution for the next dye removal test without accounting for any powder loss.

### 2.5. Cr(VI) Remediation Tests

The Cr(VI) abatement capability was tested in water with two different Cr(VI) concentrations, 10 and 50 ppm, and without any direct irradiation. 50 mg of sample was suspended in 20 mL of a K_2_Cr_2_O_7_ aqueous solution for 20 minutes and then recovered by centrifugation. After that, the sample was resuspended in other 20 mL of Cr(VI) solution for a consecutive chromium-removal test without any regeneration treatment: this cycle was repeated five times. The remaining Cr(VI) concentration was determined by UV-vis spectroscopy at the wavelength of maximum absorption (540 nm): before the analysis, 10 mL of the solution retrieved by centrifugation was treated with 200 μL of a 0.5% diphenylcarbazide (DFC) solution and 100 μL of a 0.1 M H_2_SO_4_ solution and let at rest for 15 min.

The total chromium content (hexavalent and trivalent) in solution was evaluated by atomic absorption spectroscopy (AAS). The analyses were carried out on a PinAAcle 900 T spectrometer (Perkin Elmer, Waltham, MA, USA) using air-acetylene flame and working with a wavelength of 357.87 nm. 

## 3. Results

In order to better understand the reaction pathway and tailor the materials properties to the desired application, the role of TiO_2_ and H_2_O_2_ ratios with respect to aniline dimer are herein investigated. With this aim, seven different samples were prepared as reported in Table 1. It should be noted that, when the reaction is carried out in the absence of TiO_2_, hardly any material could be produced [20].

### 3.1. Thermogravimetric Analyses

In agreement with the literature regarding doped PANI thermal degradation, TG analyses performed in air of all composites showed three main weight loss stages [22,23,24,25] (Figure 1): the first at around 100 °C is attributed to the loss of physisorbed water, the second one (150–200 °C) is due to the loss of the dopant (HCl), and the third one, which accounts for most of the weight loss, starts at around 350 °C and is attributed to the polymer backbone degradation. At 700 °C, PANI is fully degraded in all samples, therefore the remaining fraction can be attributed to TiO_2_ [26]; in this way, we could evaluate the actual composition of the samples (Table 1, columns 4 and 5).

By comparing the samples prepared with the same stochiometric amount of TiO_2_, such as 1 × 1, 1 × 2 and 1 × 3, we could expect the same %TiO_2_ in these composites, but this is not the case: increasing the amount of H_2_O_2_ leads to a higher PANI content. This trend can be also observed by comparing 2 × 1with 2 × 2 sample (34%w and 52%w of PANI, respectively) and 3 × 1 with 3 × 3 sample (27%w vs. 43%w of PANI). These results support a promoting role of H_2_O_2_ on PANI formation.

Differences in the thermal stability of the composites can be better appreciated by comparing the weight loss derivative (DTG) curves (Figure 1). The composites with higher TiO_2_ content showed the same DTG profile as the reference 1 × 1 sample (Figure 1A), where water desorption, HCl loss and polymer degradation give rise to peaks at 70, 187 and 510 °C, respectively [22,27,28,29]. Similar results are observed when both the TiO_2_ and H_2_O_2_ amounts are increased. On the other hand, when only the H_2_O_2_ amount is increased, a second peak component with higher thermal stability is appreciable for the polymer chain degradation (1 × 2 and 1 × 3 in Figure 1B): this second component becomes more important and progressively shifted to higher temperature by increasing the H_2_O_2_ quantity (625 and 640 °C for 1 × 2 and 1 × 3, respectively). Moreover, the DTG curve of the 1 × 3 sample presents also a peak at 267 °C, which could be attributed to HCl bound in a stronger way to the polymer backbone or to oligomer chains degradation. It is noteworthy that the DTG curve of PANI_ref (Appendix A) displays broader peaks with respect to the 1 × 1 curve, both in the case of dopant loss and of the polyaniline chain degradation. In particular, DTG peak components at higher temperatures are appreciable, similarly to the case of the 1 × 2 and 1 × 3 curves.

The thermal stability of PANI is known to be affected by several factors, such as the structure of the polymer backbone and oxidation state [22,30,31,32]. In the literature, the presence of interactions between PANI chains and TiO_2_ surface is reported to weaken the polyaniline inter-chain interactions, thus lowering the thermal stability of the composite [23,24,27,30,31,33,34]. This can explain the lower thermal stability of our composites with respect to PANI_ref. However, when the TiO_2_ : H_2_O_2_ ratio is weighted in favor of H_2_O_2_ (as in 1 × 2 and 1 × 3 samples), the thermal stability becomes comparable with PANI_ref, supporting the occurrence of weakened polymer-TiO_2_ interactions, possibly due to different composite formation mechanisms (vide infra).

### 3.2. Structural Properties

The structural properties of all the composites were investigated via XRPD analyses (Figure 2): all the diffractograms showed the peaks of both TiO_2_ (anatase and rutile phases) and polyaniline.

The reference composite (1 × 1) presented sharp and well resolved peaks of the ES-I form of the emeraldine salt [35], characteristic of a high crystallinity degree. The presence of well defined peaks due to crystalline PANI phases, which are far less resolved in both PANI_ref [14] and in PANI from oxidative aniline polymerization [35], is indicative of a more ordered arrangement of the polymer chains in the composite samples with respect to pure PANI. It should be noted that our composites are far more crystalline than PANI composites prepared via traditional oxidative polymerization in the presence of TiO_2_ or even other oxides [28,36,37,38,39,40]. This observation supports the occurrence, in the present case, of a different reaction mechanism promoted by the photocatalytic properties of the oxide semiconductor under UV irradiation.

By increasing both the amounts of photocatalyst and H_2_O_2_ (Figure 2A), the polymer XRPD peaks decrease in relative intensity with respect to TiO_2_ reflections: this can be only partly explained by the lower PANI content (Table 1). Moreover, particularly in the 3 × 3 composite, PANI peaks loose resolution, suggesting a lower crystallinity degree of the polymer. 

Samples 2 × 1 and 3 × 1 (i.e., with increased amount of only TiO_2_) showed similar diffractograms, with PANI peaks less intense than 1 × 1 (in agreement with their lower PANI content, Table 1), but still clearly resolved (Figure 2B).

Conversely, the composites with increased amount of only H_2_O_2_ (1 × 2 and 1 × 3) displayed less defined and overlapped PANI peaks with respect to the 1 × 1 sample (Figure 2C). In particular, in the sample 1 × 3, the fine structure of the polyaniline peaks is nearly lost, indicating a more amorphous arrangement of the polymer chains as also supported by the clearly appreciable amorphous halo.

In accordance with TGA results, increasing the amount of TiO_2_ does not significantly affect the composite properties, whereas when the TiO_2_ : H_2_O_2_ ratio is decreased, a notable effect on the PANI crystallinity is observed.

### 3.3. Morphological Properties

N_2_ adsorption isotherms under subcritical conditions (*T* = -196 °C) were acquired to evaluate the specific surface area of all samples by the BET method (Table 1). It is noteworthy that all samples show a much larger surface area with respect to PANI_ref (3 m^2^ g^−1^). In this respect, the acquired surface area of each composite was compared to the surface area of a hypothetical mechanical mixture with the same composition, calculated as the weighted sum of S_BET_ values of TiO_2_ (50 m^2^ g^−1^) and of PANI_ref [42]. All calculated values are lower than the experimental ones, with the only exceptions of 3 × 3 and 1 × 3 composites. On these grounds, we can assert that the composites are not simple mechanical mixtures, but the observed synergistic effect on morphology supports an active role of TiO_2_ in the polymer formation. It is worth noting that the two exceptions are represented by the samples with the highest H_2_O_2_/aniline dimer molar ratios. 

The variation of the composite surface area with the H_2_O_2_/aniline dimer molar ratio is reported in Figure 3. As observed in the case of XRPD analyses, the variation in the photocatalyst amount does not significantly affect the properties of the composites: in fact, samples 1 × 1, 2 × 1 and 3 × 1 have comparable surface areas (blue stars in Figure 3). On the contrary, when the H_2_O_2_ quantity is increased, the specific surface area shows a clear decreasing trend. While the surface area lowering is less marked for the 2 × 2 and 3 × 3 composites (red triangles in Figure 3), the BET area drastically drops to few square meters (similarly to PANI_ref, 3 m^2^ g^−1^) when only the H_2_O_2_ amount is increased (green dots in Figure 3). 

Therefore, as it was for crystallinity and thermal stability, the surface area is influenced mainly by the H_2_O_2_ amount while its effect can be partially compensated by the simultaneous increase of TiO_2_ amount.

Figure 4 reports the SEM images of four PANI-TiO_2_ composites: 1 × 1, 3 × 3, 3 × 1 and 1 × 3.

In agreement with BET results, the morphology of the materials is mainly influenced by the H_2_O_2_ amount used for their synthesis. In fact, 1 × 1 (Figure 4A) and 3 × 1 (Figure 4C) samples exhibit a similar sponge-like nanostructured morphology characterized by nanorod morphological structures, as clearly shown in Figure 4A inset. On the contrary, the 1 × 3 sample (Figure 4D) is characterized by larger and smooth aggregates, whereas the 3 × 3 sample displays intermediate morphological characteristics (Figure 4B). It should be noted that the reference sample (PANI_ref) presents an even more compact morphology characterized by micrometric and smooth particles [14].

### 3.4. FTIR Spectroscopy

FTIR spectra (Appendix A) confirm the successful formation of PANI in all composite samples. The ratio between C=C bands of quinoid and benzenoid rings of PANI (at 1570 and 1489 cm^−1^, respectively) is consistent with the presence of PANI mainly in the emeraldine form, in agreement with XRPD results. Other characteristic bands are the C-N bending vibration in aromatic amines at about 1300 cm^−1^, the in-plane and out-of-plane bending modes of C-N at 1030 and 875 cm^−1^, and the stretching mode of N=Q=N (Q = quinoid ring) at around 1000 cm^−1^ [43]. Moreover, the broad band in the 3400–1800 cm^−1^ region is indicative of highly conjugated structures [44]. It is noteworthy that no signals are appreciable in the 3600–3700 cm^−1^ region characteristics of oxide surface hydroxyl groups [45]. In this respect, Appendix A compares the FTIR spectrum of the composite with the highest TiO_2_ content (3 × 1, about 70%w) with those of PANI_ref and pristine TiO_2_, better highlighting the absence of the oxide surface hydroxylation in the composite. Conversely, when a conventional oxidative polymerization is adopted for the synthesis of PANI-TiO_2_ composites [12], the oxide surface hydroxylation is clearly visible in FTIR spectra. The absence of the oxide surface hydroxylation band in our composites suggests a complete coverage of the TiO_2_ particles by polyaniline chains, supporting a strong interaction between the oxide surface and the polymer (see XPS results in Section 3.7). 

The band at 1240 cm^−1^ is related to the C-N^+^ stretching vibration in the polaron structure of conducting PANI [44]. A high degree of electron delocalization is also testified by the strong band at around 1140 cm^−1^, described as “electron-like band” of PANI [46], that can be observed for all composites. 

At wavenumbers lower than 900 cm^−1^, the broad band of TiO_2_ (Appendix A) is predominant in composite samples, making difficult to appreciate overlapping bands characteristic of PANI, such as the deformation vibration modes for the aromatic rings at 741 and 689 cm^−1^ or that at 573 cm^−1^, typical of 1,4 di-substituted benzene. The TiO_2_ band in the 900–400 cm^−1^ region becomes more intense for samples with higher TiO_2_ content, in good agreement with TGA results. 

### 3.5. UV-vis Spectroscopy

UV-vis spectra of samples are reported in Appendix A. According to the literature [14,47], spectra acquired in both DMF solution (Appendix A) and acidified DMF (Appendix A) show the typical bands of polyaniline in its emeraldine form, as confirmed by the band at 430 nm, assigned to polaron–*π** transitions. 

More in detail, in Appendix A the band at 320 nm can be assigned to the *π*→*π** transition of benzenoid rings (H1), whereas that at 585 nm is due to the benzenoid to quinoid excitonic transition (H2), shifted at 890 nm when the spectra are collected in acidic conditions (Appendix A). The intensity ratio of the two bands represents an index of the oxidation state of the polymer. For PANI_ref, the H1/H2 ratio was determined to be 2.65 (Appendix A). Figure 5 and Appendix A show the dependence of the H1/H2 ratio from the H_2_O_2_/aniline dimer molar ratio. 

The results indicate that the use of a large amount of H_2_O_2_ during the composite synthesis increases the benzenoid/quinoid ratio, to the detriment of the ideal 1:1 ratio of the emeraldine form. On the other hand, the amount of photocatalysts plays no significant role in this regard. 

### 3.6. Mass Spectra

Mass spectra of the DMF-soluble fractions of 1 × 1, 1 × 2, 1 × 3 and 3 × 1 samples are reported in Figure 6; the relative MS spectrum of PANI_ref sample is reported for the sake of comparison. Although only the soluble fractions, hence short chains, are responsible for acquired data, such information can be precious to investigate the polymerization pathways.

When the reaction is carried out in the presence of a large amount of H_2_O_2_ (PANI_ref, 1 × 3 and 1 × 2), an intense peak at 366 m/z is observed, which can be attributed to linear tetramers (Appendix A) [48]. This is the only peak appreciable in the PANI_ref spectrum, whereas the MS spectra of 1 × 2 and 1 × 3 samples present three other peaks at 727, 743 and 1092 m/z, which can be assigned to linear octamers and linear dodecamers [49]. By increasing the TiO_2_/H_2_O_2_ ratio (1 × 1 and 3 × 1 samples), the peak of linear tetramers (366 m/z) disappears replaced by other peaks, e.g. at 434 m/z. As for the 1 × 1 spectrum, the 567 m/z peak can be attributed to sodium adducts of oxidized hexamers [49], while peaks at 434 and 811 m/z can be explained by the presence of hydroxylated chains (Appendix A), as also supported by XPS (see Section 3.7). Moreover, the MS spectrum of the 3 × 1 sample shows other peaks, such as those at 406 and 378 m/z, which could be explained by structures formed by consecutive C=O removal from aromatic rings, causing ring contraction (Appendix A), possibly due to radical induced reactions.

### 3.7. Study Of The Oligomerization Step

Drawing information about the reaction mechanism from the final polymer is difficult due to its low solubility, the complexity of its long chain interactions and spatial orientation. The present synthetic approach offers an opportunity to investigate the early stages of PANI synthesis, as the polymerization step occurs only upon H_2_O_2_ addition. 

In the oligomerization step, the photocatalytic effect of TiO_2_ seems to play a main role as the same synthetic procedure performed without UV irradiation leads to composites with only minor PANI content [20]. Therefore, the first stage of the reaction (before H_2_O_2_ addition) was investigated via FTIR and XPS analyses on two samples: the first one was prepared by stopping the synthetic procedure after 135 min of UV irradiation (labeled as UV), the second one was obtained in the same way but without UV irradiation (NoUV). The pristine oxide was adopted as reference.

FTIR spectra of UV and NoUV samples (Appendix A) present the same bands at around 1590, 1505, 1325 and 1160 cm^−1^ besides a broad band from 3600 to 2000 cm^−1^. The latter is indicative of a more extended conjugation with respect to the dimer molecules, whereas in the 1600-1100 cm^−1^ region the peak positions are intermediate between those of the dimer and the polyaniline. Below 800 cm^−1^ the intense and broad signal related to TiO_2_ is predominant. The clear differences between UV and NoUV spectra and those of aniline dimer and 1 × 1 composite support, in the former, the growth of oligomeric species on the TiO_2_ particle surface, as also shown by the violet color of the suspension before H_2_O_2_ addition, typical of PANI oligomers. 

On the other hand, XPS analyses presented notable differences between the two samples. The sample surface composition, as determined from survey spectra, is reported in Appendix A. Both the UV and noUV samples show similar surface composition, with a larger carbon content for the noUV composite. 

In the Ti 2p region (Figure 7A), both UV and NoUV samples showed two components: the component at lower B.E., which is the only one present in pristine TiO_2_, is attributed to Ti(IV), also on the ground of the spin-orbit splitting value of 5.7 eV; the second component at higher B.E. has been previously reported [50] also in the case of PANI-TiO_2_ composites [51]. This shifted component is prevalent in the UV sample (72%), though it has a high relative weight also in the NoUV sample (49%) (Table 2). In the literature, aniline and diazobenzene adsorption on TiO_2_ anatase and rutile single crystal surfaces has been reported to lead to a positive shift of titania bands upon direct interaction of the N atom of the organic molecule with the titania surface [52,53]. Although the here reported shift is more pronounced, it can be related to the surface adsorption of aniline dimer and of oligomer species, which thus seems to be favored by UV-irradiation.

In the C 1s region, pristine TiO_2_ displays the characteristic components due to adventitious carbon [45]. Spectra of both NoUV and UV samples showed a main component at 284.7 ± 0.1 eV which can be attributed to C-C and C-H species [54], and a second major component at ca. 286 eV due to more oxidized carbon, which can be related to the C-N/C=N bonds (Table 2) [55,56]. It is noteworthy that the UV composite shows a third component at higher B.E. (288.4 eV), which could be attributed to highly oxidized carbon bonded with oxygen [45] (Figure 7B). The latter attribution is also supported by O 1s region (Appendix A) and closely reflects MS findings.

In the O 1s region, all three samples present a main component at around 530 eV attributable to bulk lattice oxygen (Ti-O-Ti) [45,50]. A second more oxidized component at around 531 eV can be appreciated in all three samples and attributed to surface oxygen species: in the pristine TiO_2_ sample, this component can be attributed to hydroxyl groups [45,50], also on the ground of FTIR spectra (Appendix A). The lattice oxygen relative abundance is higher in pristine TiO_2_ (85%) than in NoUV and UV samples (57% and 53%, respectively): this observation can be referred to signal attenuation in the latter samples, due to the adsorbed organic species at the TiO_2_ surface. As in the case of the C 1s region, the UV sample shows a third more oxidized component (532.9 eV), accounting for about 8% of total oxygen species (Table 2), which can be attributed to oxidized carbon species (Appendix A) [56].

The N 1s region of UV and NoUV samples (Appendix A) displays a broad and less intense peak: for these reasons we preferred not to fit the N 1s signal to avoid possible misleading interpretations. In any case, a clear shift towards higher B.E. is appreciable for the irradiated sample indicative of overall more oxidized *N* species.

### 3.8. Dye Removal Tests

The dye sorption capability of all composites was tested towards methyl orange (MO) removal in water medium: MO was chosen as model pollutant for anionic azo-dyes due to its bio-accumulation risk and eco-toxicity [57]. Most of the samples exhibit very good performance in terms of dye removal (>95% for tests with 50 ppm MO starting concentration, Appendix A). A closer look at the data shows a main role of the composite specific surface area, as the best performing samples have areas larger than 30 m^2^ g^−1^, in good agreement with the literature [14,20]. The 1 × 3 sample gave rise to the worst performance (barely 48% of dye removed), which could be related to its very low surface area and compact morphology. It should be noted that even this sample outperforms PANI_ref (MO removal 10%) [20].

However, surface area is not the only factor at play. For instance, by comparing the 3 × 3 and 1 × 2 composites, the sample with the larger surface area (24 vs. 21 m^2^ g^−1^, respectively) exhibited lower performance in terms of MO removal (respectively 85% and 92%). This result could be related to the higher PANI content of the 1 × 2 composite as it is well known that polyaniline sorption mechanism is based on both an anion-exchange process (Cl^-^ vs. dye) and short-range interactions, such as hydrogen bonding and *π*-*π* stacking [20,58,59]: thus, composites with a more accessible polymer structure and more sorption sites are characterized by better dye-sorption performances. 

Samples retain good dye removal performance in a broad concentration range (10-200 ppm), displaying a linear increase in the dye sorbed amount (Appendix A). A stress test on a representative composite (1 × 1) was performed by reusing the material several times for the remediation of concentrated dye solutions (50 ppm) without any regeneration in between consecutive runs. The first two sorption tests show an almost complete dye removal, while afterwards the dye removal capacity starts to decrease as expected (Figure 8). Consecutive runs were carried out until an almost complete loss of the sorption properties was observed. A maximum sorption capacity of 62 mg g^−1^ was determined, which favorably compares with literature results concerning the adsorption of acid red dyes by PANI-based materials [60,61] and activated carbon [62]. It should be noted that even after the maximum sorption capacity was reached, the initial sorption properties of the composite could be fully restored by simply immersing it in alkaline water for 24 h (20 mL of NaOH solution at pH around 10). Moreover, this regeneration approach enables to recover unaltered the sorbed dye, opening the door to its reusage in a circular economy perspective.

For the sake of comparability with real-life conditions, sorption tests were carried out also in simulated drinking water (Appendix A). The sorption test exhibits comparable results with respect to the dye solution in the absence of electrolytes (94% vs. 96%, respectively). Moreover, in simulated drinking water, the sorption properties could be retained for many more consecutive runs leading to a notable increase in the total MO sorbed amount: after eleven reuses without regeneration, a 94 mg g^−1^ dye-sorption capacity was measured. It should be noted that the stress test in simulated drinking water was stopped well before the sample sorption limit could be reached (Figure 8). In simulated drinking water, besides the dye molecules, there are several and more concentrated ions in solution, including chlorides. Such ions could lead to competitive adsorption at the polyaniline charged sites and the higher ionic strength could, on one hand, partially shield the dye molecules reducing their ability to interact with the PANI chains, while on the other hand, it could promote sites dissociation. In the pristine PANI composite, positively charged sorption sites are fully compensated by Cl^-^ counterions, which are mainly exchanged with the anionic dye during the first sorption test. In the subsequent cycles, dye molecules and chloride ions in solution compete for displacing adsorbed species (either dye molecules or chlorides) on the grounds of electrostatic interactions. All these competitive interactions cause a decrease in the percentage of removed MO, while postponing the sorbent saturation.

### 3.9. Chromium Remediation Tests

Prompted by its optimal dye-removal performances, the 1 × 1 composite was also tested for the remediation of Cr(VI) in solution. Hexavalent chromium is a well-known toxic, carcinogenic and mutagenic species; on the other hand, its trivalent form, Cr(III), is about 1000 times less hazardous and can be removed by adsorption or precipitation (Cr(OH_3_)) at neutral pH [63]: thus, most of the Cr(VI) abatement methods are based on its reduction to Cr(III) followed by the removal of the latter. However, the chemical reduction process consumes large quantities of reductants and generates high amounts of secondary waste. Polyaniline has been reported to reduce Cr(VI) to Cr(III) and to remove both the species by sorption [63]: both the sorption and reduction mechanisms are strongly pH dependent. The best sorption performances are achieved at pH 5, while the reduction is reported to be favored by strong acidic conditions (pH < 2) [63].

In the present work, tests were carried out at pH 1.8 using two different Cr(VI) concentrations (10 and 50 ppm), well within the range of chromium concentration in industrial wastewaters [64]. In both cases, the 1 × 1 composite was able to reduce > 99% of the Cr(VI) and to directly remove 50% of the total chromium in solution in just 20 minutes. Furthermore, the present composite outperforms PANI_ref in terms of both Cr(VI) reduction and total chromium removal results; as a matter of fact, the latter reduces only 30% of Cr(VI) and removes merely 2% of the total chromium in a 10 ppm solution.

Stress tests were carried out also in this case: after retrieval by centrifugation, the 1 × 1 composite was reused for five consecutive removal tests without any intermediate regeneration treatment. In order to simulate experimental conditions close to real life applications, stress tests were carried out without adjusting the pH value. Five consecutive cycles were carried out using either 10 ppm and 50 ppm Cr(VI) solutions at spontaneous pH (Figure 9). The 1 × 1 composite was able to maintain over 90% Cr(VI) reduction even at the fifth reuse at the highest adopted chromium concentration, reaching a total Cr(VI) reduction capacity of 98 mg/g. This performance favorably compares with previous literature reports [63,65,66,67]; in this respect, it should be noted that the cycle test was stopped well before the sample sorption limit could be reached. 

Total chromium removal markedly increases during the cycles: from values close to 50% during the first test to over 96% at the fifth reuse. This behavior can be related to the progressive spontaneous increase of pH in the suspension (from pH 1.8 for the first tests to values around 5 in the last tests). As a matter of fact, the sorption mechanism of cations is favored with pH values around 5, while it is hindered at pH ≤ 2 due to the charge repulsion between the positively charged polyaniline and Cr(III) cations. Concerning anions removal, such as Cr(VI) mainly present in solution as HCrO_4_^-^, two adsorption mechanisms can be proposed. The first one is based on a dopant change, whereas the second mechanism involves electrostatic interactions among the anion and the positive iminic groups of the polymer. It should be noted that the present composite displays unaltered Cr(VI) reduction capacity in a broad range of pH values, which enables to modulate the sorption performance at will. When a complete abatement of all chromium species is required, pH values around 5 can be adopted. On the contrary, if Cr(III) species are to be separately recovered (as required in a circular economy perspective), strong acidic conditions are the optimal choice.

These results demonstrate that the synthesized composite materials are characterized by a fast and efficient Cr(VI) removal capacity, adjustable chromium sorption properties, as well as a high degree of reusability, even in the total absence of regeneration treatments. Moreover, preliminary tests have shown that a quick washing with a 0.05 M HCl solution can be adopted to reactivate the composites.

## 4. Discussion

The main characteristic of the synthetic pathway herein proposed is the clear distinction between the oligomerization and polymerization steps. This two-step reaction mechanism points towards different roles played by TiO_2_/UV with respect to H_2_O_2_, which are both essential for the PANI formation. The first step involves the adsorption of aniline dimers at the oxide surface, as shown by both FTIR and XPS spectra exhibiting a loss of surface hydroxylation and a notable second component in the Ti 2p spectra. The latter effect is more marked in the presence of UV irradiation, in accordance with the literature reporting an increased omniphilicity of light-irradiated TiO_2_ [68]. This initial step is crucial to promote the final PANI crystallinity and porous morphology, which can be related to the controlled and oriented growth of the oligomer chains at the oxide surface, explaining the unique crystallinity degree of our samples with respect to literature composites [28,36,37,39,40]. 

Light irradiation is fundamental to promote the dimer-TiO_2_ interaction as well as the formation of radicals via photocatalytic reactions, which can act as initiators of the oligomerization step. In this respect, XPS and MS spectra consistently show the occurrence of radical-assisted oxidative processes in the UV irradiated sample, as also supported by literature studies about the photocatalytic degradation of aniline by TiO_2_ [69,70,71]. 

It should be underlined that the reaction cannot proceed further without the addition of H_2_O_2_, as clearly shown by the change in color from violet to dark green, characteristic of polyaniline, upon H_2_O_2_ addition. Oxygen peroxide is known to act as both a radical source initiating PANI synthesis, as well as an oxidant supporting the polymer growth [72]. In PANI_ref synthesis, previous studies have shown that H_2_O_2_ dissociation in the presence of a catalyst can produce radical species (HO·) able to promote the formation of initiators (aniline dimer radicals) [72]. These are crucial for the first part of the reaction, which follows a radical mechanism. Afterwards, H_2_O_2_ acts as an oxidant during the growth of polymer chains [73]. However, it is still unclear whether the more oxidized form of polyaniline (pernigraniline) is initially produced and then spontaneously decays to the more stable form of emeraldine, or whether the more reduced form of PANI (leucoemeraldine) is immediately synthesized and then gradually oxidized to emeraldine [74]. In the present study, characterizations confirmed that the formation of active species (aniline dimer radicals) takes place on the TiO_2_ surface in the first part of the reaction, thus guaranteeing the formation of PANI oligomers. Hence, in our hypothesis, hydrogen peroxide plays mainly the role of the oxidant during polymer growth. 

The H_2_O_2_ amount plays a major role on the final materials properties, promoting on one side an increase in the PANI content, which comes at the cost of a lower crystallinity, altered benzenoid/quinoid ratio, and worsened morphological properties. In this respect, the PANI_ref synthesis involves a homogeneous catalysis and a much higher H_2_O_2_ : aniline dimer ratio, leading to poor structural and morphological features. Here 5-times lower H_2_O_2_ : aniline dimer ratios can be successfully employed for the synthesis due to the synergistic role played by H_2_O_2_ and TiO_2_/UV. It is hard to assert whether H_2_O_2_ promotes only the growth of adsorbed chains on the oxide surface or elicits reactions in homogeneous phase, as *e.g.* in PANI_ref. However, on the grounds of TGA and MS findings, we can suggest that, at low H_2_O_2_ amounts, the growth is driven by the adsorbed oligomers. At increasing H_2_O_2_ quantities, evidence of a parallel homogeneous pathway becomes instead appreciable. For instance, TGA curves of 1 × 2 and 1 × 3 samples are consistent with weaker polyaniline-TiO_2_ interactions and MS spectra present soluble components in common with the PANI_ref spectrum, at variance with those shown by the 1 × 1 sample.

The insight gained into the relative roles of TiO_2_/UV and H_2_O_2_ on the reaction mechanism allows us to modulate the composite properties in a broad range. The nanocomposite composition, crystallinity, morphology and conjugation degree can all be modified by working on the reagent ratios to suit the desired application. In the present case, the composite performance could be enhanced mainly by increasing the specific surface area using low H_2_O_2_ : aniline dimer ratios. In this respect, the TiO_2_ content in the nanocomposite could be increased without affecting the materials characteristics, a much desired feature for applications directly exploiting the TiO_2_ semiconductor properties. On the other hand, increasing H_2_O_2_ amount during synthesis leads to poorer structural and morphological properties. However, even at the highest H_2_O_2_ : aniline dimer ratios tested in the present work, the resulting composites displayed promoted crystallinity and surface areas with respect to purely homogeneous synthetic pathways (PANI_ref and literature composites), enabling us to tune the PANI content in the nanocomposite to a wider extent (up to 75%w). 

## 5. Conclusions

PANI-TiO_2_ nanocomposites prepared via UV-light mediated green procedure were here studied and applied to pollutant removal in water. The preparation of PANI-TiO_2_ composites has been quite extensively reported in the literature by using both preformed particles and TiO_2_ sols/gels [12,29,37,38,75,76]. However, the present synthesis differs from the state-of-the-art as the photocatalytic activity of TiO_2_ is directly exploited to initiate a green polymerization process, leading to materials with unique properties. Here, we varied the H_2_O_2_ and TiO_2_ amounts to demonstrate their different but essential roles on the composite thermal, optical, structural and morphological properties. The first reaction step, driven by TiO_2_ and UV irradiation, was found to be a radical process which promotes the final composite crystallinity and polymer-oxide interaction. The H_2_O_2_ addition is needed for PANI preparation and its amount has a crucial role on the reaction pathway. Low H_2_O_2_ quantities can preferentially promote the growth of oligomer chains adsorbed on TiO_2_ particles, giving rise to highly porous, large surface area nanocomposites with good crystallinity but lower PANI-content. On the other hand, high H_2_O_2_ amounts seem to promote a homogenous polymer formation process, which leads to nanocomposites with higher PANI content and thermal stability, at the expense of their crystallinity and porous morphology. However, even at the highest tested H_2_O_2_ amount, the present nanocomposites display characteristic structural and morphological features that differentiate them from PANI samples from homogenous catalysis reaction (PANI_ref) as well as from composites prepared by conventional oxidative polymerization routes. These results enabled us to tune the composite properties in order to adapt them to different applications. In the present case, materials prepared with low H_2_O_2_ amounts showed the best performance in terms of pollutant removal capability, exhibiting a fast and complete MO and Cr(VI) abatement in tests on a broad concentration range. In the literature PANI-TiO_2_ composites are mostly applied as photocatalyst for pollutant degradation [29,38,75,76], with the inherent limitations of this approach such as slow reaction kinetics, interfering effect of common electrolytes and possible accumulation of stable and toxic by-products. In this respect, the prepared nanocomposites show instead excellent sorption properties that can be used even in the presence of concentrated electrolytes (e.g., chlorides, bicarbonates), enabling the recovery of the pristine molecule upon mild treatments. Moreover, in the case of Cr(VI) abatement, the metal ion reduction is here obtained without the need of additional reagents (as opposed to the case of photocatalysis in which a hole scavenger is needed) and with the concurrent sorption of Cr(III). Furthermore, the composite showed excellent reusability, even in the total absence of regeneration treatments and under real-life conditions, as shown by the good performance in simulated drinking water. Future work will investigate the immobilization of the nanocomposite on macroscopic devices in order to simplify the recovery at the end of the treatment.

## Figures and Tables

**Figure 1 nanomaterials-10-00441-f001:**
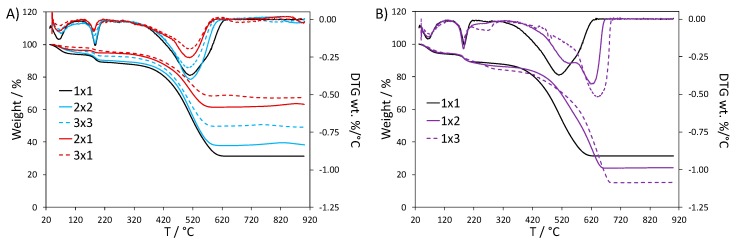
TGA and DTG curves of PANI-TiO_2_ nanocomposites: (**A**) effect of TiO_2_ quantity and (**B**) effect of H_2_O_2_ amount.

**Figure 2 nanomaterials-10-00441-f002:**
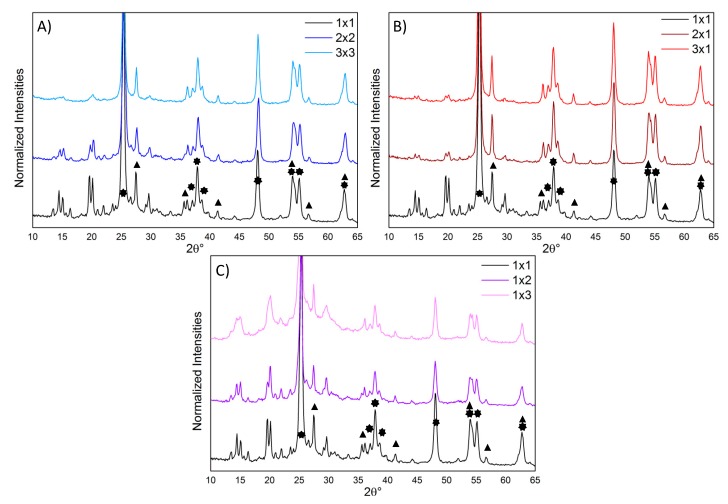
Normalized XRPD diffractograms (with respect to the anatase TiO_2_ (213) peak): (**A**) effect of changing both the TiO_2_ and H_2_O_2_ amounts; (**B**) effect of changing only the TiO_2_ amount; (**C**) effect of changing only the H_2_O_2_ amount. The stars and triangles highlight the peaks related to the TiO_2_ crystalline phases (anatase and rutile, respectively [41]).

**Figure 3 nanomaterials-10-00441-f003:**
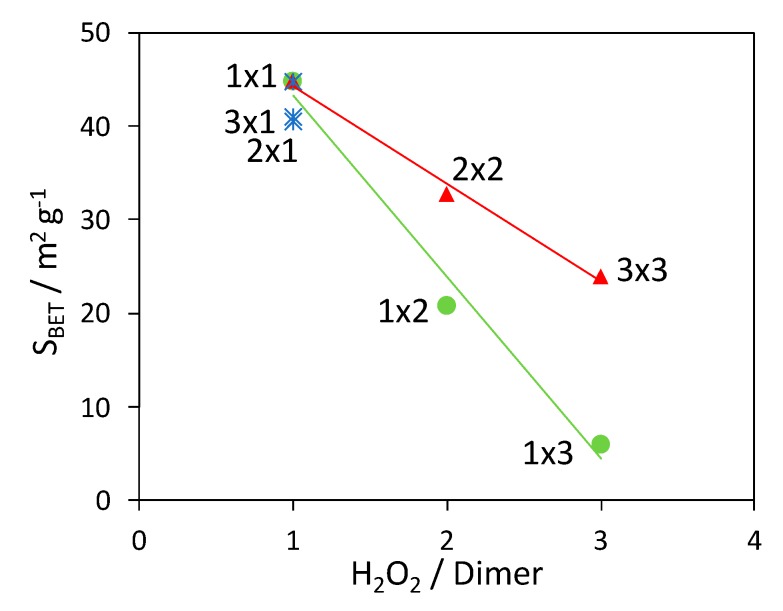
Dependence of the composite surface area vs. the H_2_O_2_/aniline dimer molar ratio: effect of changing the H_2_O_2_ amount (green dots), TiO_2_ content (blue stars), and of varying both parameters (red triangles). Lines are added as a guide for the eye.

**Figure 4 nanomaterials-10-00441-f004:**
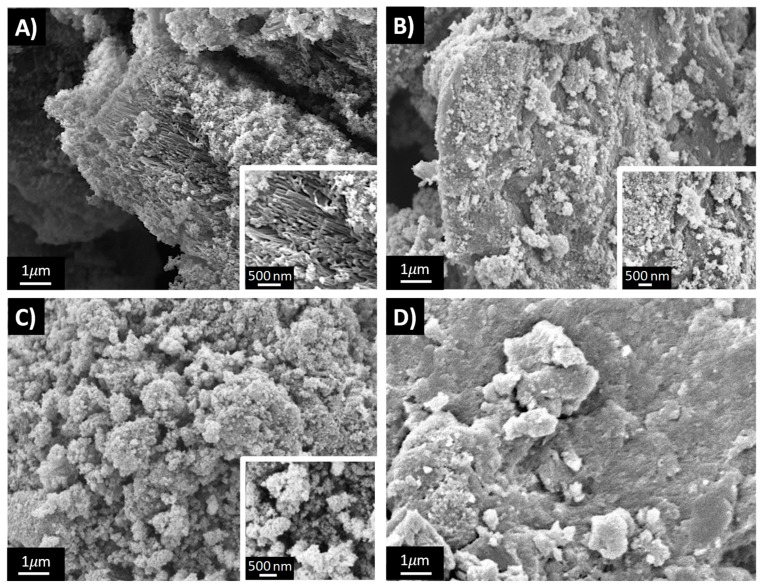
SEM images of: (**A**) 1 × 1, (**B**) 3 × 3, (**C**) 3 × 1, (**D**) 1 × 3.

**Figure 5 nanomaterials-10-00441-f005:**
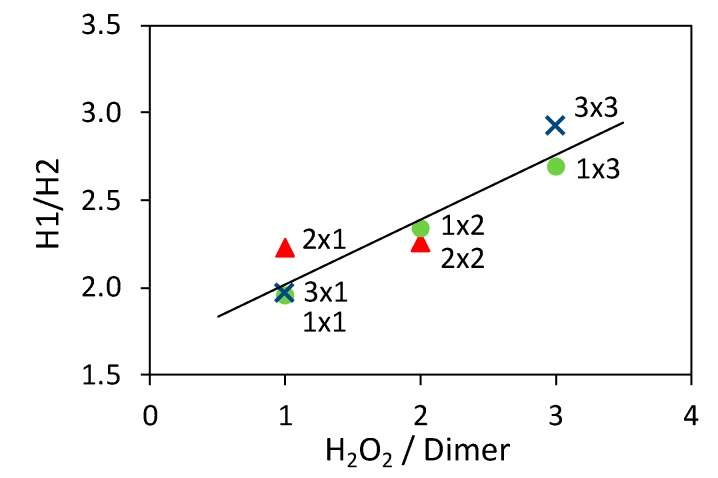
Dependence of the H1/H2 ratio from the H_2_O_2_/aniline dimer molar ratio. Samples prepared using the same TiO_2_ amount are represented with the same marker and the line is added as guide for the eyes.

**Figure 6 nanomaterials-10-00441-f006:**
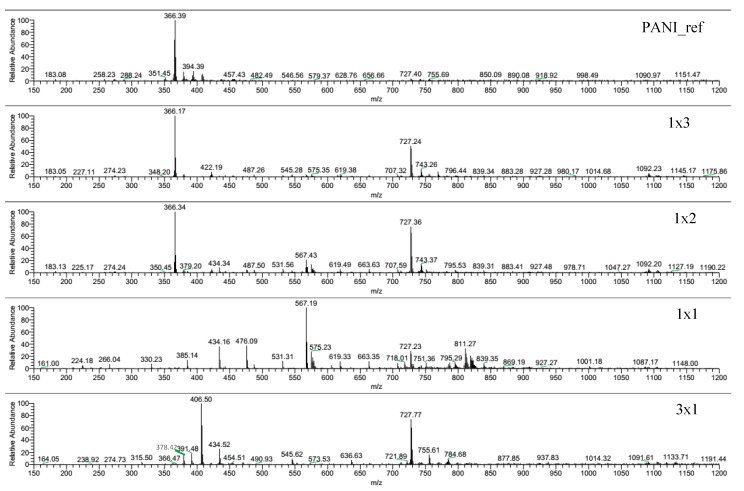
MS spectra of DMF-soluble fractions of PANI_ref, 1 × 3, 1 × 2, 1 × 1, and 3 × 1 samples.

**Figure 7 nanomaterials-10-00441-f007:**
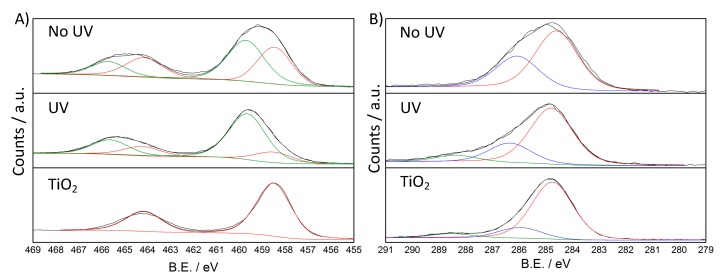
High resolution XPS spectra, and relative fitting, of NoUV, UV and TiO_2_ samples: (**A**) Ti 2p region and (**B**) C 1s region.

**Figure 8 nanomaterials-10-00441-f008:**
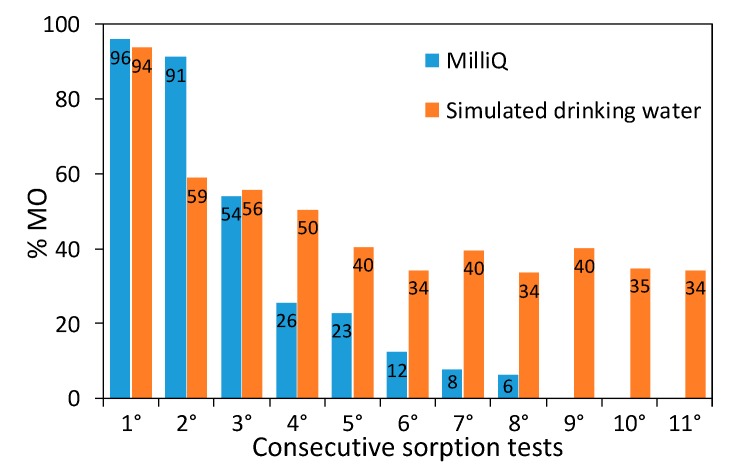
MO removal during sorption tests: consecutive runs of the 1 × 1 sample performed in both Milli-*Q* water and simulated drinking water.

**Figure 9 nanomaterials-10-00441-f009:**
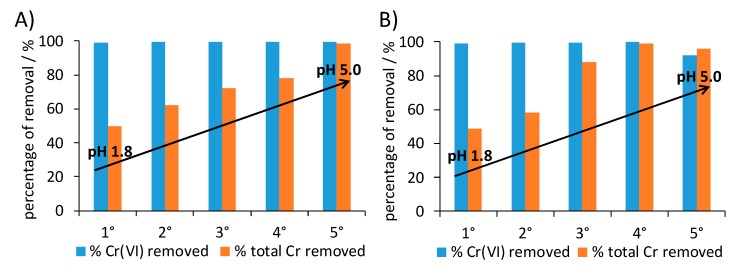
Consecutive chromium removal tests for 1 × 1 nanocomposite: (**A**) 10 ppm and (**B**) 50 ppm initial Cr(VI) concentration. For each test, the removal of Cr(VI) (± 0.04%) and of total chromium (± 4%) are reported.

**Table 1 nanomaterials-10-00441-t001:** Stoichiometric molar ratios of photocatalyst and H_2_O_2_ with respect to aniline dimer (columns 2 and 3), weight percentage of PANI and TiO_2_ in the final products obtained from TG analyses (columns 4 and 5), measured specific surface area of each composite and calculated surface area of a physical mixture with same composition (columns 6 and 7).

Sample	Molar Ratios		Sample Composition^1^		*S*_BET_ (m^2^ g^−1^)	*S*_BET_ Mix ^2^ (m^2^ g^−1^)
	TiO_2_/Dimer	H_2_O_2_/Dimer	%PANI	%TiO_2_		
1 × 1	0.4	1.0	58	31	44.8	17.2
2 × 2	0.8	2.0	52	38	32.8	20.6
3 × 3	1.2	3.0	43	50	24.0	26.3
2 × 1	0.8	1.0	34	61	40.5	31.5
3 × 1	1.2	1.0	27	69	41.0	35.3
1 × 2	0.4	2.0	64	24	20.8	13.9
1 × 3	0.4	3.0	75	15	6.0	9.8

^1^ The sum of %PANI and %TiO_2_ is lower than 100 because of water and dopant content. ^2^ Surface area of theoretical mechanical mixtures with the same composition of the samples, calculated as the weighted sum of PANI_ref and TiO_2_ surface area, taking into account the sample composition determined from TGA.

**Table 2 nanomaterials-10-00441-t002:** B.E. and peak area of the components of the UV and NoUV samples compared to pristine TiO_2_.

Region	TiO_2_		NoUV		UV	
	Position (eV)	% Area	Position (eV)	% Area	Position (eV)	% Area
Ti 2p	458.5	100	458.5	51	458.5	28
	-		459.7	49	459.7	72
C 1s	284.7	81	284.6	65	284.8	68
	285.9	14	286.1	35	286.4	24
	288.4	5	-		288.4	8
O 1s	529.7	85	529.8	57	530.1	53
	531.4	15	531.2	43	531.2	39
	-		-		532.9	8

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
