# Peer review of "Photocatalytic and Oxidative Synthetic Pathways for Highly Efficient PANI-TiO2 Nanocomposites as Organic and Inorganic Pollutant Sorbents"

_nanomaterials, 2020, doi:10.3390/nano10030441_

Round 1
Reviewer 1 Report
The manuscript 727570 from Cionti et al is about the preparation of polyaniline with an eco-friendly procedure initiated by UV-irradiated TiO2. The nanocomposites prepared by varying the TiO2 : H2O2 : aniline-dimer molar ratios and characterized for their thermal, optical, morphological, structural and surface properties. The authors studied their application as sorbents for the removal of Cr(VI) and methyl orange.
The manuscript is well written, with a lot of work and helpful discussion. It can be accept for publication after minor revision
The main comment is about the discussion of the SSA of the samples and the correlation with the preparation method and the sorbent ability. I believe that the authors should discuss this more, as the SSA is an important parameter, although it is difficult to find a trend. Maybe a figure similar to Fig. 4 may help with H2O2;dimer as x or the ratio of PANI/TiO2
Also, please discuss in more details, if possible, the role of oxidant in the preparation route.
Line 90 Please change x with × and in all other points used.
Figure 2 (XRD patterns) are complicated and a better figure is needed. I suggest the authors to move some patterns in y axes up or down to be easy to read and to distinguish the peaks of each sample.
Table in Figure 4 and Figure 7 should be deleted
Reviewer 2 Report
The paper presents some interesting data on the synthesis of PANI-TiO2 nanocomposites via a "green" methodology, as claimed by the authors.
The following comments must be taken into account in the revision of the manuscript:
A greener synthesis of PANI is based on electrochemical oxidative condensation of aniline, which does not require any hydrogen peroxide or solvents. This should be reflected in the introduction. The aniline dimer used in the paper as a precursor is usually produced by a chemical method that cannot be qualified as a green technology, therefore the methodology proposed in the manuscript is based on a non-green precursor and this reduces the greenness of the proposed technology The authors used the term "nanocomposite" for the materials they synthesized, but there are no proofs in the manuscript of the nano-size nature of the material, except for surface area values that are not quite high to give the grounds to belive that the materials are really nanomaterials. The particle sizes must be determined and the particle size distribution should be presented in the manuscript. I would recommend to resubmit this paper to MDPI "Materials" journal
Reviewer 3 Report
The manuscript entitled ".Photocatalytic and oxidative synthetic pathways for highly efficient PANI-TiO2 nanocomposites as organic and inorganic pollutant sorbents." proposes a study on an alternative synthesis to conventional methods for the preparation of PANI using TiO2 and H2O2 avoiding, according to the authors, the use of toxic carcinogenic compounds.
Finally, the work continues with a characterization of the synthesized material and with particular tests that involve the removal of organic molecules such as orange methyl, taken as the representative of organic pollutants, and the removal of Cr (IV) as the representative of inorganic pollutants.
From the data reported, the authors state that the material studied has a high removal efficiency with the possibility of reusing it without the need for regeneration.
The topic of the study is interesting, although it is often confusing and unclear and difficult to read. I therefore believe that before it can be considered for its publication it must be revisited in some parts.
Here are some suggestions for the authors:
- (paragraph 2.1 Sample preparation). This part is difficult to read. I believe it is necessary at least to insert a table reporting all the studied samples, the molar systems used and the respective acronyms. This paragraph should be improved.
- (line 139). The initial MO concentration of the solution is reported to be 50 ppm, while in line 397 it is stated that "Sample retain good dye removal performance in a broad concentration range (10-200ppm)". This part is unclear and should be rewritten. How many MO solutions have been used.
- (Line 140) What is the pH value.
- (Figure 2) The profiles of the different samples are not distinguishable.
- (Paragraph 3.8 Dye removal Tests). As it can be explained that the material in the first cycles does not lead to a total abatement of the dye, but in the subsequent cycles and without its regeneration it can still reduce 50%. For example, in the 3rd cycle it cuts 56%, almost as a demonstration of a saturation of absorption, but in the 4th it is able to reduce 50% again without its regeneration. I believe it is appropriate to justify this aspect
- (Figure 8). It would be appropriate to add the corresponding pH values in the graphs.
- How do you plan to regenerate the material?
- Have studies and research been done to confidently state that the proposed synthesis for this particular nanocomposite can be considered eco-friendly?
Round 2
Reviewer 2 Report
My comments have been properly addressed in the revision
Reviewer 3 Report
I believe that the manuscript has been improved and in the present form it can be considered for its publications.
With best regards